# *Mycobacterium vaccae* NCTC 11659, a Soil-Derived Bacterium with Stress Resilience Properties, Modulates the Proinflammatory Effects of LPS in Macrophages

**DOI:** 10.3390/ijms24065176

**Published:** 2023-03-08

**Authors:** Evan M. Holbrook, Cristian A. Zambrano, Caelan T. O. Wright, Elizabeth M. Dubé, Jessica R. Stewart, William J. Sanders, Matthew G. Frank, Andrew S. MacDonald, Stefan O. Reber, Christopher A. Lowry

**Affiliations:** 1Department of Integrative Physiology, University of Colorado Boulder, Boulder, CO 80309, USA; 2Department of Molecular, Cellular, and Developmental Biology, University of Colorado Boulder, Boulder, CO 80309, USA; 3Department of Psychology and Neuroscience, University of Colorado Boulder, Boulder, CO 80309, USA; 4Lydia Becker Institute of Immunology and Inflammation, University of Manchester, Manchester M13 9NT, UK; 5Laboratory for Molecular Psychosomatics, Department of Psychosomatic Medicine and Psychotherapy, Ulm University Medical Center, 89081 Ulm, Germany; 6Center for Neuroscience and Center for Microbial Exploration, University of Colorado, Boulder, CO 80309, USA; 7Department of Physical Medicine and Rehabilitation and Center for Neuroscience, University of Colorado Anschutz Medical Campus, Aurora, CO 80045, USA; 8Veterans Health Administration, Rocky Mountain Mental Illness Research Education and Clinical Center (MIRECC), The Rocky Mountain Regional Veterans Affairs Medical Center (RMRVAMC), Aurora, CO 80045, USA; 9Military and Veteran Microbiome Consortium for Research and Education (MVM-CoRE), Aurora, CO 80045, USA; 10Senior Fellow, inVIVO Planetary Health, of the Worldwide Universities Network (WUN), West New York, NJ 07093, USA

**Keywords:** IL-10, IL-12, IL-23, immunoregulation, macrophage, monocyte, *Mycobacterium vaccae*, real-time RT-PCR, TGF-β1, THP-1

## Abstract

Inflammatory conditions, including allergic asthma and conditions in which chronic low-grade inflammation is a risk factor, such as stress-related psychiatric disorders, are prevalent and are a significant cause of disability worldwide. Novel approaches for the prevention and treatment of these disorders are needed. One approach is the use of immunoregulatory microorganisms, such as *Mycobacterium vaccae* NCTC 11659, which have anti-inflammatory, immunoregulatory, and stress-resilience properties. However, little is known about how *M. vaccae* NCTC 11659 affects specific immune cell targets, including monocytes, which can traffic to peripheral organs and the central nervous system and differentiate into monocyte-derived macrophages that, in turn, can drive inflammation and neuroinflammation. In this study, we investigated the effects of *M. vaccae* NCTC 11659 and subsequent lipopolysaccharide (LPS) challenge on gene expression in human monocyte-derived macrophages. THP-1 monocytes were differentiated into macrophages, exposed to *M. vaccae* NCTC 11659 (0, 10, 30, 100, 300 µg/mL), then, 24 h later, challenged with LPS (0, 0.5, 2.5, 250 ng/mL), and assessed for gene expression 24 h following challenge with LPS. Exposure to *M. vaccae* NCTC 11659 prior to challenge with higher concentrations of LPS (250 ng/mL) polarized human monocyte-derived macrophages with decreased *IL12A*, *IL12B*, and *IL23A* expression relative to *IL10* and *TGFB1* mRNA expression. These data identify human monocyte-derived macrophages as a direct target of *M. vaccae* NCTC 11659 and support the development of *M. vaccae* NCTC 11659 as a potential intervention to prevent stress-induced inflammation and neuroinflammation implicated in the etiology and pathophysiology of inflammatory conditions and stress-related psychiatric disorders.

## 1. Introduction

Inflammatory conditions, such as allergic asthma, and conditions in which inflammation is considered a risk factor, e.g., stress-related psychiatric disorders including anxiety disorders, mood disorders, and trauma- and stressor-related disorders, such as posttraumatic stress disorder (PTSD) [1,2], are prevalent and are a significant cause of disability worldwide [3,4]. One strategy to reduce the risk of these conditions would be to use anti-inflammatory or immunoregulatory approaches to mitigate inflammatory responses [5,6]. As posited by the “Hygiene” hypothesis or “Old Friends” hypothesis, industrialized nations have higher rates of chronic low-grade inflammation, inflammatory diseases, and psychiatric disorders due to urbanization resulting in decreased exposures to immunoregulatory microorganisms (i.e., microorganisms that promote a balanced expression of regulatory and effector T cells) [7,8]. Indeed, growing up on a farm provides protection against allergic asthma [8], while healthy young persons raised in a city, without daily exposure to pets for the first 15 years of life, respond to a psychosocial stressor with exaggerated increases in numbers of circulating peripheral blood mononuclear cells (PBMCs) and increases in circulating concentrations of interleukin 6 (IL-6), a proinflammatory cytokine, relative to those who were raised on farms in close proximity to farm animals [9].

One mechanism that has been implicated in exaggerated inflammation in the context of both allergic asthma and stress-related psychiatric disorders is the recruitment of inflammatory monocytes to the airways or central nervous system [1,10,11,12]. Monocyte-derived macrophages in house dust mite-allergic mice or persons with asthma have an inflammatory phenotype, characterized by increased IL-6 and IL-12 production [13]. Likewise, studies in rodents have shown that IL-6 is increased in individuals that respond to psychosocial stress with a susceptible phenotype following subsequent chronic stress exposure. Indeed, individual variability in ex vivo stimulated IL-6 from leukocytes prior to stress exposure predicts stress susceptibility versus resilience [14,15]. Furthermore, the transfer of hematopoietic progenitor cells from stress-susceptible mice to IL-6-deficient mice was sufficient to transfer a stress-susceptible phenotype, highlighting the importance of bone marrow-derived leukocytes to psychosocial stress-induced anxiety-like defensive behavioral responses and depressive-like responses [15]. In particular, monocytes were elevated in individuals that developed a susceptible phenotype after stress exposure, and monocyte numbers were negatively correlated with behavioral responses [15,16]. In clinical populations, circulating monocyte numbers, relative abundance, or inflammatory gene expression patterns in monocytes have been associated with risk of anxiety disorders [17], mood disorders [18,19,20,21], and suicide risk [18,19,22], and have been identified among top-ranking features predicting future PTSD symptoms [23]. Furthermore, monocytes from persons with depression show an altered response to lipopolysaccharide (LPS) [24].

Studies in persons with a diagnosis of depression have shown elevated circulating concentrations of other proinflammatory cytokines, including IL-12, a heterodimer between IL-12p35 and IL-12p40, encoded by *IL12A* and *IL12B*, respectively. For example, individuals with depression have elevated plasma IL-12 concentrations relative to healthy controls [25,26,27,28], which are decreased after treatment [25,26,27]. Likewise, the ratio of IL-12 to anti-inflammatory cytokines, such as transforming growth factor β1 (TGF-β1), is decreased following antidepressant treatment [26]. Indeed, meta-analyses of inflammatory markers in depression reveal increases in a diverse range of proinflammatory cytokines, including IL-12 [29,30]. Finally, treatment with tumor necrosis factor (TNF) and IL-12/IL-23 blockers can improve depressive symptoms in persons with inflammatory disease [31].

One strategy, therefore, is to develop interventions that restore adequate anti-inflammatory and immunoregulatory responses and, thus, reduce the risk of immune-mediated conditions and stress-related psychiatric disorders, in which inflammation is a risk factor. One of these immunoregulatory interventions is *Mycobacterium vaccae (M. vaccae)* NCTC 11659, which has anti-inflammatory, immunoregulatory, and stress-resilience properties (for review, see [5]. However, little is known about how *M. vaccae* NCTC 11659 affects specific immune cell targets, including monocyte-derived macrophages that can drive inflammatory and neuroinflammatory responses and, thus, increase risk of immune-mediated conditions and stress-related psychiatric disorders [1,16,32]. Monocytes that differentiate into macrophages with a proinflammatory bias [1], or exosomes derived from these cells [33,34], can infiltrate the central nervous system (CNS) and induce an inflammatory neuroimmune environment, which is a risk factor for the development of stress-related psychiatric disorders, including anxiety disorders, mood disorders, and trauma- and stressor-related disorders, such as PTSD [1,32]. However, macrophages with an anti-inflammatory bias that infiltrate the CNS are vital in CNS repair [35,36] and might be key in suppressing inappropriate inflammation in the CNS. In line with recommendations to avoid the use of the terms “resting” and “activated” innate immune cells, and dichotomous “M1-like” and “M2-like” categorization of these cells [37,38,39,40,41], here we simply refer to proinflammatory or anti-inflammatory bias, or polarization of macrophages, and describe the specific changes in gene expression.

This study used a whole-cell, heat-killed preparation of *M. vaccae* NCTC 11659 [IMM-201]. *M. vaccae* NCTC 11659 is a saprophytic environmental bacterium with anti-inflammatory and immunoregulatory properties. The original strain, designated R877, was isolated by Dr. John L. Stanford and Roger C. Paul (University College London and Middlesex Hospital Medical School, London), from mud in the Kyoga Nile valley near Lake Kyoga in Uganda [42]. Multiple single-colony isolations were conducted by John Stanford and John M. Grange to clone a stable rough variant of *M. vaccae* R877, designated R877R. The rough variant was deposited as *M. vaccae* NCTC 11659 and a heat-killed preparation of *M. vaccae* NCTC 11659 was initially evaluated in clinical studies of leprosy and tuberculosis (for review, see [5]. Subsequent studies in mice demonstrated that immunization with *M. vaccae* NCTC 11659 provided long-term protection from allergic airway inflammation in a manner dependent on induction of regulatory T cells (Treg) and production of anti-inflammatory cytokines, including IL-10 and TGF-β1 ([43]; for review, see [5,44]. We have since described in detail the effects of immunization with a heat-killed preparation of *M. vaccae* NCTC 11659 to prevent: (1) stress-induced increases in spontaneous colitis; (2) stress-induced exaggeration of secretion of proinflammatory cytokines, including IL-6, from mesenteric lymph node cells treated with anti-CD3 antibody ex vivo; and (3) stress-induced increases in anxiety-like defensive behavioral responses [45,46,47,48,49,50,51,52,53,54,55], including effects relevant to three of the four major diagnostic criteria for PTSD (see Appendix A in [45]; see Table 1 in [56].

We set out to determine if *M. vaccae* NCTC 11659 could shift human monocyte-derived macrophages to a less inflammatory phenotype that would respond to subsequent immune challenges with LPS with a bias toward anti-inflammatory responses, characterized by increased ratios of genes encoding anti-inflammatory cytokines, i.e., IL-10 and TGF-β1, to proinflammatory cytokines, i.e., IL-12 and IL-23 [57,58]. For a list of abbreviations for all gene symbols, see Appendix A.

## 2. Results

### 2.1. Cell Viability

Cell viability was assessed using flow cytometry gating of live singlets based on forward and side scatter (FSC/SSC), which revealed no effect of either *M. vaccae* NCTC 11659 or LPS on cell viability (see Appendix A). Sample sizes are listed in Appendix A. This was supported by an assessment of *ACTB* mRNA expression using real-time reverse transcription quantitative-polymerase chain reaction (real-time RT-PCR), which revealed no consistent effects of either *M. vaccae* NCTC 11659 or LPS on cell viability as assessed by the expression of this reference gene (see Section 2.3 below).

### 2.2. Effects of Heat-Killed M. vaccae NCTC 11659 (300 µg/mL) and LPS (250 ng/mL) on Gene Expression Using NanoString nCounter^®^ Inflammation v2 Panel

To determine if there was any effect of treatment condition on reference genes, we analyzed counts of the six reference genes standardized using negative and positive control probes (i.e., *CTLC*, *GAPDH*, *GUSB*, *HPRT1*, *PGK1*, and *TUBB*). Two-way ANOVA revealed main effects for *M. vaccae* NCTC 11659, LPS, or interactions between these factors for each of the six reference genes (see Appendix A). Thus, all normalization was done using Removing Unwanted Variation-III (RUV-III), specifically designed to address this issue in the NanoString nCounter gene expression assay [59].

#### 2.2.1. Effects of LPS

Previous studies have shown that LPS (*Escherichia coli* 0111:B4) is a potent inducer of inflammation in human monocyte-derived macrophages [60]. After 24 h of exposure to 250 ng/mL LPS, 107 of the 249 inflammation-related genes in the CodeSet (i.e., 43%) were differentially expressed (*p* < 0.05 and fold change either less than –1.5 or greater than 1.5) within the BBS vehicle groups between LPS conditions (i.e., BBS/250 ng/mL LPS vs. BBS/RPMI 1640 vehicle). Of these 107 differentially expressed genes, 57 were upregulated and 50 were downregulated (Appendix A). A detailed table of all differentially expressed genes can be found in Appendix A.

Overall, the response to LPS was consistent with previous studies showing that the majority of genes responsive to LPS in THP-1-derived macrophages contain known or putative NF-κB sites, including *CCL3*, *CCL4*, *CXCL1*, *CXCL2*, *CXCL8* (*IL8*), *CXCL10*, *CXCR4*, *IL1RN* [61]. As LPS signals through the toll-like receptor 4 (TLR4)-MD-2 complex [62], the differentially expressed genes found in the Toll-Like Receptor Signaling KEGG pathway, including TLR4, can be found in Appendix A.

#### 2.2.2. Effects of *M. vaccae* NCTC 11659

Previous studies have reported that *M. vaccae* NCTC 11659, in the absence of subsequent stress exposure or LPS challenge, has an “adjuvant-like” effect in vivo, for example, increasing bronchopulmonary *Il1b*, *Il6*, and *Tnf* mRNA expression 12 h to 3 days following intratracheal administration [63], increasing secretion of IL-6 from mesenteric lymph node cells stimulated with anti-CD3 antibody in vitro [46], and increasing hippocampal *Il6* mRNA expression [45]. After 48 h of exposure to 300 µg/mL *M. vaccae* NCTC 11659, 120 of the 249 endogenous genes in the CodeSet (i.e., 48%) were differentially expressed (*p* < 0.05 and fold change either less than –1.5 or greater than 1.5) within the RPMI 1640 vehicle groups between *M. vaccae* NCTC 11659 conditions (i.e., 300 µg/mL *M. vaccae* NCTC 11659/RPMI 1640 vehicle vs. BBS/RPMI 1640 vehicle). Of these 120 differentially expressed genes, 68 were upregulated and 52 were downregulated (Figure 1A). A detailed table of all differentially expressed genes can be found in Appendix A. *M. vaccae* NCTC 11659 increased *IL10*, which suppresses NFκB signaling in human monocytes [64,65]. Notably, *M. vaccae* NCTC 11659 has been shown previously to signal through activation of human TLR2 [66] and here exposure to *M. vaccae* NCTC 11659 upregulated *RAC1*, involved in the suppression of NF-κB signaling following activation of TLR2/TLR1 or TLR2/TLR6, which in turn would be expected to downregulate the expression of *BCL2*, *BCL2A1*, *BCL2L1*, *BIRC2*, *BIRC3*, *CCL4*, *CCL4L1*, *CCL4L2*, *CFLAR*, *CXCL1*, *CXCL2*, *CXCL3*, *CXCL8* (IL-8), *GADD45A*, *GADD45B*, *GADD45G*, *IL1B*, *IL-6*, *IL-12A*, *IL-12B*, *NFKB2*, *NFKBIA*, *PTGS2*, *TNF*, *TNFAIP3*, *TRAF1*, *TRAF2*, *VCAM1*, and *XIAP*. Indeed, exposure to *M. vaccae* NCTC 11659 downregulated a large number of genes downstream of NFκB signaling, including *BCL2L1*, *BIRC2*, *CCL3*, *CXCL3*, *IL12B*, *PTGS2*, *TNF*, *TNFAIP3*, and *TRAF2* responses to the subsequent challenge with LPS (see below). *OASL*, downstream of STAT1 and STAT2 signaling, was also strongly downregulated (Figure 1A). As *M. vaccae* NCTC 11659 signals through toll-like receptor 2 (TLR2) [66], the differentially expressed genes found in the Toll-Like Receptor Signaling KEGG pathway, including TLR2, can be found in Appendix A.

#### 2.2.3. Effects of *M. vaccae* NCTC 11659 in LPS-Challenged Cells

Previous studies have shown that immunization with *M. vaccae* NCTC 11659 attenuates stress-induced exaggeration of inflammatory gene expression in vivo [45,46]. Following forty-eight hours of exposure to 300 µg/mL *M. vaccae* NCTC 11659 or vehicle and 24 h of exposure to 250 ng/mL LPS, 115 of the 249 genes in the CodeSet (i.e., 46%) were differentially expressed (*p* < 0.05 and fold change either less than −1.5 or greater than 1.5) within the LPS groups between *M. vaccae* NCTC 11659 conditions (i.e., 300 µg/mL *M. vaccae* NCTC 11659/250 ng/mL LPS vs. BBS/250 ng/mL LPS). Out of these 115 differentially expressed genes, 52 were upregulated and 63 were downregulated (Figure 1B). A detailed table of all differentially expressed genes can be found in Appendix A. Consistent with a downregulation of *TNF* and *CREB1, MAP2K4, MAP3K5, RIPK1, TRADD,* and *TRAF2* (involved in transcriptional responses to TNF signaling) in *M. vaccae* NCTC 11659 exposed, LPS-challenged macrophages, a number of genes downstream of TNF signaling were also downregulated, including *CCL2*, *CXCL10*, *CSF1*, *PTGS2*, and *TNFAIP3*. In contrast, genes involved in IL-10/STAT3 signaling were upregulated, including *IL10* and *IL10RB*, while IL-10 responsive genes, e.g., *IL1A*, *IL12B*, *IRF7*, *STAT1* and *TNF* [67,68], were downregulated. This is consistent with previous studies showing that, at the transcriptional level, IL-10 attenuates expression of a subset of genes activated by TLR signaling, accounting for 20–25% of mRNAs induced after LPS stimulation including *Il1a*, *Il12b*, *Il18,* and *Tnf* [69]. The effects of *M. vaccae* NCTC 11659 in LPS-challenged cells on differentially expressed genes found in the Toll-Like Receptor Signaling KEGG pathway can be found in Appendix A.

#### 2.2.4. Effects of LPS in *M. vaccae* NCTC 11659-Exposed Cells

When the effects of LPS were evaluated among *M. vaccae* NCTC 11659-exposed cells, i.e., *M. vaccae* NCTC 11659/LPS challenge versus *M. vaccae* NCTC 11659/RPMI 1640 exposed cells, 148 of the 249 genes in the CodeSet (i.e., 59%) were differentially expressed (*p* < 0.05 and fold change either less than –1.5 or greater than 1.5) between the LPS groups within *M. vaccae* NCTC 11659 conditions (i.e., 300 µg/mL *M. vaccae* NCTC 11,659/250 ng/mL LPS vs. 300 µg/mL *M. vaccae* NCTC 11659/RPMI 1640). Of these 148 differentially expressed genes, 65 were upregulated and 83 were downregulated (Appendix A). A detailed table of all differentially expressed genes can be found in Appendix A. Among *M. vaccae* NCTC 11659-exposed cells, the LPS challenge resulted in the downregulation of a number of genes involved in proinflammatory responses, particularly NF-κB responsive genes including *BCL2L1*, *BIRC2*, *IL-12B*, *NFKB2*, *NFKBIA*, *TNF*, *TNFAIP3*, and *TRAF2*. The effects of LPS in *M. vaccae* NCTC 11659-exposed cells on the differentially expressed genes found in the Toll-Like Receptor Signaling KEGG pathway can be found in Appendix A.

#### 2.2.5. Effects of Heat-Killed *M. vaccae* NCTC 11659 (30 µg/mL) and LPS on Gene Expression Using NanoString nCounter^®^ Inflammation Panel

For a description of the effects of exposure of THP-1-derived macrophages to a lower concentration of *M. vaccae* NCTC 11659 (30 µg/mL) (see Appendix A.

#### 2.2.6. Summary of NanoString nCounter Inflammation Panel

As data from the NanoString nCounter Inflammation Panel suggested that *M. vaccae* NCTC 11659 upregulated *IL10* expression and downregulated *IL12B* expression in LPS-challenged macrophages, and the anti-inflammatory effects of *M. vaccae* NCTC 11659 in vivo have been shown to depend on IL-10 and TGF-β1 signaling, we followed up in detail to assess the effects of *M. vaccae* NCTC 11659 and LPS on *IL10*, *TGFB1*, *IL12A*, *IL12B*, and *IL23A* using real-time RT-PCR.

### 2.3. Validation of Effects of Heat-Killed M. vaccae NCTC 11659 and LPS on IL12A, IL12B, IL23A, IL10, and TGFB1 mRNA Expression Using Real-Time RT-PCR

Here we describe results from experiments designed to measure the expression of genes encoding major inflammatory cytokines in macrophages by extracting total RNA from THP-1 monocyte-derived macrophages and performing real-time PCR as described above. Analysis of the housekeeping gene, *ACTB*, for differential expression across treatment groups found no meaningful differences in expression (Appendix A). Overall, similar effects of *M. vaccae* NCTC 11659 and LPS were observed on *IL12A*, *IL12B*, *IL23A*, *IL10*, and *TGFB1* mRNA expression using real-time RT-PCR as in *Experiment 1* (Appendix A).

### 2.4. Effects of Heat-Killed M. vaccae NCTC 11659 and LPS on IL10:IL12A mRNA Expression Ratio

Expression levels of *IL12A*, *IL12B*, *IL23A*, and *IL10* are important in their own right to characterize the immune environment, but the ratio between anti-inflammatory and inflammatory cytokines in the environment is an important feature of macrophage polarization [70]. For example, high expression of IL-12 and IL-23, but low expression of IL-10, is characteristic of macrophages with proinflammatory polarization [57,58]. In contrast, high expression of IL-10, but low expression of IL-12, is characteristic of macrophages with an anti-inflammatory polarization, particularly macrophages implicated in anti-inflammatory and immunoregulatory effects [57,58]. This suggests that the ratios of *IL10* to *IL12A*, *IL12B,* and *IL23A* may be meaningful measures of macrophage polarization. The ratio of *IL10* to *IL12A* expression was calculated in THP-1 monocyte-derived macrophages after incubating them with various concentrations of *M. vaccae* NCTC 11659 and LPS as described below. In line with the low expression of *IL12A* mRNA in the vehicle (0 ng/mL LPS) condition, 70% of *IL10:IL12A* mRNA expression ratio data were real-time PCR non-detects (Appendix A). Analysis of the *IL10:IL12A* mRNA expression ratio revealed an *M. vaccae* NCTC 11659 × LPS interaction (*F*_(12, 102)_ = 2.461, *p* < 0.01); there was no main effect of *M. vaccae* NCTC 11659 (*F*_(4, 102)_ = 1.51, *p* = 0.21), but there was a significant main effect of LPS (*F*_(3, 102)_ = 27.20, *p* < 0.001). In BBS/vehicle-treated THP-1 monocyte-derived macrophages, *IL10* mRNA expression was approximately three-fold greater than *IL12A* mRNA expression (Figure 2A). Exposure of THP-1 monocyte-derived macrophages to higher concentrations of *M. vaccae* NCTC 11659 in the presence of 0.5 ng/mL LPS decreased the *IL10:IL12A* mRNA expression ratio (Figure 2A), whereas exposure of THP-1 monocyte-derived macrophages to the highest concentration of *M. vaccae* NCTC 11659 (300 µg/mL) in the presence of 250 ng/mL LPS increased the *IL10*:*IL12A* mRNA expression ratio (Figure 2A). Exposure of THP-1 monocyte-derived macrophages to LPS in the absence of *M. vaccae* NCTC 11659 did not affect the *IL10:IL12A* mRNA expression ratio (Figure 2A). Post hoc *p* values for planned pairwise comparisons can be found in Appendix A.

#### 2.4.1. *IL10:IL12B* mRNA Expression Ratio

In line with data from the NanoString nCounter platform, both *M. vaccae* NCTC 11659 and LPS had pronounced effects on *IL10:IL12B* mRNA expression in THP-1 monocyte-derived macrophages. The ratio of *IL10* to *IL12B* expression was calculated in THP-1 monocyte-derived macrophages after incubating them with various concentrations of *M. vaccae* NCTC 11659 and LPS as described above. Analysis of the *IL10:IL12B* mRNA expression ratio revealed an *M. vaccae* NCTC 11659 × LPS interaction (*F*_(12, 120)_ = 14.19, *p* < 0.001) as well as main effects of *M. vaccae* NCTC 11659 (*F*_(4, 120)_ = 23.92, *p* < 0.001) and LPS (*F*_(3, 120)_ = 660.7, *p* < 0.001). Exposure of THP-1 monocyte-derived macrophages to *M. vaccae* NCTC 11659 in the absence of LPS and in the presence of 0.5 ng/mL LPS decreased the *IL10:IL12B* mRNA expression ratio in a concentration-dependent manner (Figure 2B). Likewise, exposure of THP-1 monocyte-derived macrophages to LPS in the absence of *M. vaccae* NCTC 11659 decreased the *IL10:IL12B* mRNA expression ratio in a concentration dependent manner. In contrast, at the higher concentrations of LPS (2.5 ng/mL and 250 ng/mL) pretreatment with *M. vaccae* NCTC 11659 increased the *IL10:IL12B* mRNA expression ratio (Figure 2B). Post hoc *p* values for planned pairwise comparisons can be found in Appendix A.

#### 2.4.2. *IL10:IL23A* mRNA Expression Ratio

The ratio of *IL10* to *IL23A* expression was calculated in THP-1 monocyte-derived macrophages after incubating them with various concentrations of *M. vaccae* NCTC 11659 and LPS as described above. Analysis of the *IL10:IL23A* mRNA expression ratio revealed an *M. vaccae* NCTC 11659 × LPS interaction (*F*_(12, 139)_ = 8.57, *p* < 0.001) as well as main effects of *M. vaccae* NCTC 11659 (*F*_(4, 139)_ = 23.15, *p* < 0.001) and LPS (*F*_(3, 139)_ = 438.9, *p* < 0.001). Exposure of THP-1 monocyte-derived macrophages to *M. vaccae* NCTC 11659 in the absence of LPS or in the presence of low concentrations of LPS (0.5 and 2.5 ng/mL) decreased the *IL10:IL23A* mRNA expression ratio in a concentration-dependent manner (Figure 2C). Exposure of THP-1-derived macrophages to LPS in the absence of *M. vaccae* NCTC 11659 decreased the *IL10:IL23A* mRNA expression ratio in a concentration-dependent manner (Figure 2C). Pretreatment with *M. vaccae* NCTC 11659 increased the *IL10:IL23A* mRNA expression ratio at the highest concentration of LPS (250 ng/mL; Figure 2C). Post hoc *p* values for planned pairwise comparisons can be found in Appendix A.

#### 2.4.3. *TGFB1:IL12A*, *TGFB1:IL12B*, and *TGFB1:IL23A* mRNA Expression Ratios

As mentioned above, expression levels of *IL12A*, *IL12B*, *IL23A*, and *TGFB1* are important in their own right to characterize the immune environment, but the ratio between anti-inflammatory and inflammatory cytokines in the environment is an important feature of macrophage polarization. For example, high expression of IL-12, but low expression of TGF-β1, is characteristic of macrophages with a proinflammatory bias [57,58]. In contrast, high expression of TGF-β1, but low expression of IL-12, is characteristic of macrophages with an anti-inflammatory bias, particularly macrophages implicated in anti-inflammatory and immunosuppressive effects [57,58]. This suggests that the ratio of *TGFB1* to *IL12A*, *IL12B*, and *IL23A* may be a meaningful measure of macrophage polarization. The ratio of *TGFB1* to *IL12A* expression was calculated in THP-1 monocyte-derived macrophages after incubating them with various concentrations of *M. vaccae* NCTC 11659 and LPS as described above. The analysis revealed similar effects of *M. vaccae* NCTC 11659 and LPS on these ratios as the *IL10*:*IL12A*, *IL10*:*IL12B*, and *IL10*:*IL23A* mRNA expression ratios (Appendix A–C). Post hoc *p* values for planned pairwise comparisons can be found in Appendix A.

## 3. Discussion

Here we report evidence that *M. vaccae* NCTC 11659, a soil-derived bacterium with anti-inflammatory, immunoregulatory, and stress resilience properties, modulates the proinflammatory effects of LPS with *M. vaccae* NCTC 11659 inducing “adjuvant-like” effects in the absence of LPS or at low concentrations of LPS (0, 0.5 ng/mL) and mitigating the effects of LPS at higher concentrations of LPS (250 ng/mL). These data suggest that *M. vaccae* NCTC 11659 might function to induce an anti-inflammatory or immunoregulatory phenotype under hyper-inflammatory conditions, which are observed in response to allergic airway inflammation, in response to extreme trauma or stress, or in individuals with a diagnosis of trauma- and stressor-related disorders, such as PTSD [71]. *M. vaccae* NCTC 11659 alone had adjuvant-like effects, based on increases in proinflammatory cytokine mRNA expression and decreases in anti-inflammatory to proinflammatory cytokine mRNA ratios in the absence of subsequent LPS challenge, or challenge with a low concentration of LPS (0.5 ng/mL). LPS alone was a reliable stimulator of inflammation based on concentration-dependent increases in proinflammatory cytokine mRNA expression and decreased anti-inflammatory to proinflammatory cytokine mRNA expression. Interestingly, pretreatment with *M. vaccae* NCTC 11659 prior to LPS challenge resulted in attenuation of LPS-induced exaggeration of inflammation based on decreased proinflammatory cytokine mRNA levels and increased anti-inflammatory to proinflammatory cytokine mRNA ratios. Specifically, at the highest concentration of LPS (250 ng/mL), pretreatment with *M. vaccae* NCTC 11659 shifted THP-1 monocyte-derived macrophages from a proinflammatory phenotype (*IL10* low and *IL12B* high) toward an anti-inflammatory phenotype (*IL10* high and *IL12B* low).

### 3.1. M. vaccae NCTC 11659 Had “Adjuvant-like” Effects on THP-1 Monocyte-Derived Macrophages

Acute effects of *M. vaccae* NCTC 11659 on human monocyte-derived macrophages, in the absence of subsequent LPS stimulation, were largely proinflammatory when assessed 48 h later, shifting macrophages toward a proinflammatory macrophage phenotype. This is evidenced by concentration-dependent increases in *IL12B* and *IL23A* mRNA expression, hallmark cytokines of proinflammatory macrophages that together encode IL-23, a heterodimeric cytokine of IL-12p40 (encoded by *IL12B)* and IL-23p19 (encoded by *IL23A*) [58], and concentration-dependent decreases in the *IL10:IL12B*, *IL10:IL23A*, *TGFB1:IL12B*, and *TGFB1:IL23A* mRNA expression ratios [58] in human THP-1 monocyte-derived macrophages. The administration of *M. vaccae* NCTC 11659 alone, therefore, seems to act as its own adjuvant (i.e., ensuring immune activation before subsequent immune challenge). These effects are consistent with previous studies of *M. vaccae* NCTC 11659 in vivo. For example, in a murine model of allergic airway inflammation, intra-tracheal administration of *M. vaccae* NCTC 11659 increases pulmonary *Il1b*, *Il6*, and *Tnf* mRNA expression acutely, 12 h to 3 days following injection. In a mouse model of chronic psychosocial stress, administration of *M. vaccae* NCTC 11659, in single-housed control animals (i.e., not exposed to subsequent chronic psychosocial stress), *M. vaccae* NCTC 11659 increases IL-6 secretion, and the IL-6:IL-10 ratio, in freshly isolated mesenteric lymph node cells stimulated with anti-CD3 antibody in vitro, assessed over one month following the final immunization with *M. vaccae* NCTC 11659, suggesting that adjuvant-like effects of *M. vaccae* NCTC 11659 can be long-lasting [46]. Similarly, in a model of inescapable stress in rats, administration of *M. vaccae* NCTC 11659 to home-cage control animals increases hippocampal *Il6* mRNA expression, assessed one week after the final immunization [45]. Together, the in vitro data reported here are consistent with previous studies demonstrating that, in the absence of subsequent psychosocial stress, inescapable stress, or immunological challenge, *M. vaccae* NCTC 11659 has persistent adjuvant-like effects.

### 3.2. Lipopolysaccharide (LPS) Was a Reliable Stimulator of Inflammatory Cytokine mRNA Expression in THP-1 Monocyte-Derived Macrophages

As expected, exposure of human monocyte-derived macrophages to LPS, in a concentration-dependent manner, shifted macrophages toward a proinflammatory macrophage phenotype. Similar to the administration of *M. vaccae* NCTC 11659 alone, LPS stimulation in the absence of *M. vaccae* NCTC 11659 increased *IL12B* and *IL23A* mRNA expression in a concentration-dependent manner (again, characteristic of proinflammatory macrophages) and decreased the *IL10:IL12B*, *IL10:IL23A*, *TGFB1:IL12B*, and *TGFB1:IL23A* mRNA expression ratios in a concentration-dependent manner in human THP-1 monocyte-derived macrophages. LPS stimulation at the higher concentrations studied (i.e., 2.5 ng/mL and 250 ng/mL) was a reliable stimulator of inflammatory cytokine mRNA expression and may be most relevant to how psychosocial stressors or immune stimulation with LPS are reliable stimulators of inflammatory cytokine mRNA expression in human monocyte-derived macrophages, as previously demonstrated [1].

### 3.3. Treatment with M. vaccae NCTC 11659 Prior to Immune Stimulation with LPS Resulted in Attenuation of an LPS-Induced Proinflammatory Phenotype in THP-1 Monocyte-Derived Macrophages

Exposure to *M. vaccae* NCTC 11659 attenuated subsequent proinflammatory responses to immune stimulation with LPS. Unlike how *M. vaccae* NCTC 11659 stimulated a proinflammatory response in the absence of LPS challenge, *M. vaccae* NCTC 11659, in a concentration-dependent manner, attenuated proinflammatory responses to subsequent LPS challenge. This is evidenced by the attenuation of LPS-induced inflammatory responses, such as attenuating LPS-induced increases in *IL12B* and *IL23A* mRNA expression. In addition, exposure to *M. vaccae* NCTC 11659 attenuated LPS-induced decreases in the *IL10:IL12B*, *IL10:IL23A*, *TGFB1:IL12B*, and *TGFB1:IL23A* mRNA expression ratios in human THP-1-derived macrophages. These reported effects are potentially relevant for previously reported effects of *M. vaccae* NCTC 11659 to attenuate allergic airway inflammation in a murine model of allergic airway inflammation, or to attenuate stress-induced inappropriate, highly inflammatory responses. These effects are consistent with what has been described in vivo [5,43,44,45,46,72]) and may represent a form of innate immune tolerance.

### 3.4. Treatment with M. vaccae NCTC 11659 Prior to Immune Stimulation with LPS Resulted in the Promotion of an Anti-Inflammatory Phenotype and Inhibition of LPS-Induced Immune Activation in THP-1 Monocyte-Derived Macrophages

We report evidence supporting the hypothesis that *M. vaccae* NCTC 1169 induces features of macrophages involved in anti-inflammatory, immunoregulatory, and immunosuppressive effects [58,73]. *M. vaccae* NCTC 11659 did not directly upregulate *IL10* or *TGFB1* mRNA expression in cells subsequently challenged with LPS; however, it increased the mRNA expression ratios of *IL10:IL12B, IL10:IL23A*, *TGFB1:IL12B*, and *TGFB1:IL23A* upon subsequent immune stimulation with LPS. These findings have implications on how macrophages stimulated with *M. vaccae* NCTC 11659 may locally modulate T cell responses in the tissue, promoting immunoregulation directly at the site of inflammation [73]. A graphical illustration of our reported effects of *M. vaccae* NCTC 11659 and LPS on THP-1 monocyte-derived macrophages can be found in Figure 3.

### 3.5. Limitations

One limitation of our reported findings was the low expression of *IL12A* mRNA in THP-1 monocyte-derived macrophages, resulting in real-time RT-PCR non-detects (Cq > 40) that were excluded from the analysis. This resulted in low *IL12A* sample sizes, particularly in the RPMI 1640 condition, which affected the analysis of *IL12A* mRNA as well as the *IL10:IL12A* and *TGFB1:IL12A* mRNA ratios. The low *IL12A* expression in the RPMI 1640 condition is likely due to the lack of IFN-γ stimulation, an inducer of *IL12A* transcription especially important in macrophages [70,74,75]. Of course, as is common with in vitro studies, we were also limited by only specific cell types being present in the culture (only monocytes and macrophages), so cell-to-cell interactions could not directly be studied, such as T cell differentiation. Furthermore, human PMA-differentiated THP-1-derived macrophages may not accurately represent human monocyte-derived macrophages, as it has been shown that these cell types can differ in terms of LPS and IFN-γ-induced cytokine, chemokine, and growth factor secretion [76], although differences may be dependent on the specific protocol used [77]. For example, studies of mycobacterial infection have found that PMA-differentiated THP-1-derived macrophages and human monocyte-derived macrophages respond similarly in terms of bacterial uptake, viability and host response to drug-susceptible and drug-resistant mycobacterial infections [78]. Further, the THP-1 cell line used was isolated from a single person (i.e., a 1-year-old male that had acute monocytic leukemia); thus, the extent to which our reported findings can be generalized to monocyte-derived macrophages in females, or other populations, cannot be inferred from these data. In addition, our reported findings were limited by measuring gene expression at a single timepoint. Therefore, it cannot be inferred that our conclusions apply to other timepoints, especially considering the temporally dynamic functions of the genes measured [79]. Finally, it will be important to confirm that transcriptional changes observed here are associated with changes in protein synthesis and release, again, taking into account the temporal dynamics of the response.

### 3.6. Clinical Implications

Previous studies suggest that inappropriate inflammation is a clinical feature of allergic asthma and a risk factor for the development of anxiety disorders, mood disorders, and trauma- and stressor-related disorders, such as PTSD (for review, see [56]. For example, elevated biomarkers of inflammation have been identified as risk factors for the future development of PTSD following exposure to trauma [23,80], and persons with a diagnosis of PTSD have decreased Treg [81] and enhanced spontaneous production of proinflammatory cytokines from PBMCs [82]. Meanwhile, persons with a diagnosis of PTSD have increased serum concentrations of LPS, lipopolysaccharide-binding protein (LBP; a biomarker of “leaky gut”), and HMGB1, a damage-associated molecular pattern (DAMP) [71], leading the authors to suggest that “Going forward, additional investigations to evaluate the microbiota and approaches to modify the intestinal microenvironment may be a useful adjunct approach to complement existing treatments for PTSD.” The present study suggests that *M. vaccae* NCTC 11659 is a candidate for mitigating the physiological and psychological impacts of allergic asthma and stress-related psychiatric disorders, which can be highly comorbid [83,84].

### 3.7. Future Directions

Future directions include assessing the impact of *M. vaccae* NCTC 11659 on LPS activation of human primary monocytes isolated from donor PBMCs, compared to human PBMC monocyte-derived macrophages. This would enable confirmation of THP-1 work in human primary cells, with multiple donor testing possible. Future directions also include the assessment of additional genes using real-time RT-PCR, such as *ARG1*, a hallmark of macrophages with an anti-inflammatory phenotype, co-culturing THP-1 monocyte-derived macrophages with naïve T cells to directly assess the effects of *M. vaccae* NCTC 11659 on T cell differentiation and function, and assessment of the effects of *M. vaccae* NCTC 11659 on immune activation by immune challenges other than LPS. Furthermore, future studies should determine if *M. vaccae* NCTC 11659 effects are generalizable to other rapidly growing mycobacteria, or molecular constituents of *M. vaccae* NCTC 11659 that have been shown to have anti-inflammatory effects in macrophages, such as 10(*Z*)-hexadecenoic acid [85]. Likewise, it will be important to determine to what extent these results can be replicated in vivo. In mice, future studies should examine bone marrow-derived monocyte education following administration of *M. vaccae* NCTC 11659 in vivo, which could be informative in relation to understanding monocyte/innate immune cell differentiation, priming, and trained immunity, versus tolerance in the bone marrow compartment [86].

### 3.8. Conclusions

In 2013, in an article titled, “Harnessing monocyte-derived macrophages to control central nervous system pathologies: no longer ‘if’ but ‘how’” Shechter and Schwartz wrote, “Taken together, these recent advances reveal a dramatic therapeutic opportunity for controlled harnessing of macrophages for repair of the damaged CNS following acute insults, in neurodegenerative conditions, and in psychiatric disorders” [35]. We feel that the current study highlights how microbe-based interventions might, in the near term, prove to be novel and effective approaches to the realization of these dramatic therapeutic opportunities.

## 4. Materials and Methods

### 4.1. Experimental Design

#### 4.1.1. *Experiment 1*, NanoString Platform


The overall experimental design for *Experiment 1* is illustrated in Appendix A. Briefly, in *Experiment 1*, human THP-1 monocyte-derived macrophages were exposed to heat-killed preparations of *M. vaccae* NCTC 11659 (30 µg/mL or 300 µg/mL) or sterile borate-buffered saline (BBS) vehicle. Twenty-four hours later, the THP-1 monocyte-derived macrophages were challenged with lipopolysaccharide (LPS; *Escherichia coli* 0111:B4; 250 ng/mL) or RPMI 1640 vehicle. Finally, 24 h later, total RNA was extracted and assessed using the NanoString Platform. A timeline of *Experiment 1* can be found in Appendix A.

#### 4.1.2. Experimental Design: *Experiment 2*, Real-Time RT-PCR Validation

The overall experimental design for *Experiment 2* is illustrated in Appendix A. Briefly, in *Experiment 2*, human THP-1 monocyte-derived macrophages were exposed to heat-killed preparations of *M. vaccae* NCTC 11659 (10 µg/mL, 30 µg/mL, 100 µg/mL, or 300 µg/mL) or sterile borate-buffered saline (BBS) vehicle. Twenty-four hours later, the THP-1 monocyte-derived macrophages were challenged with lipopolysaccharide (LPS; *Escherichia coli* 0111:B4; 0.5 ng/mL, 2.5 ng/mL, or 250 ng/mL) or RPMI 1640 vehicle. Finally, 24 h later, total RNA was extracted and assessed using real-time RT-PCR. A timeline of *Experiment 2* can be found in Appendix A.

### 4.2. M. vaccae NCTC 11659 Preparation

After 24 h in fresh, non-PMA media, the THP-1 monocyte-derived macrophages were challenged with 30 or 300 
μg/mL concentrations (*Experiment 1*) or 10, 30, 100, or 300 
μg/mL concentrations (*Experiment 2*) of *M. vaccae* NCTC 11659 using sterile BBS to dilute from a 10 mg/mL stock (batch ENG#1, BioElpida, Lyon, France) to the desired concentration of *M. vaccae* NCTC 11659. Stocks were swirled each time before pipetting into the wells to ensure the suspension was distributed equally. *M. vaccae* NCTC 11659 treatment consisted of adding 20 µL of either sterile BBS (vehicle) or 0.26, 0.78, 2.60, or 7.80 mg/mL *M. vaccae* NCTC 11659 stocks for the 10, 30, 100, or 300 µg/mL concentrations of *M. vaccae* NCTC 11659, respectively.

### 4.3. Cell Culture and M. vaccae NCTC 11659 and LPS Exposures

THP-1 cells are a human monocyte cell line isolated from peripheral blood from a 1-year-old human male that had acute monocytic leukemia. THP-1 cells (TIB-202^TM^, American Type Culture Collection (ATCC), Manassas, VA, USA) were cultured in T75 cell culture flasks (Cat. No. 229520, CELLTREAT Scientific Products, Pepperell, MA, USA) in 5% CO_2_ at 37 °C in a humidified incubator in RPMI 1640 media (Cat. No. 10-043-CV, Corning Cellgro, Manassas, VA, USA) containing 10% fetal bovine serum (FBS; Cat. No. F9423, Sigma-Aldrich, Saint Louis, MO, USA), 5 mM 2-mercaptoethanol (Cat. No. 190242, MP Biomedicals, Irvine, CA, USA), and 100 U/mL penicillin/100 
μg/mL streptomycin (Cat. No. 15140-122, Gibco, Waltham, MA, USA). The media were replaced every 3 days through centrifugation (500× *g* for 5 min at room temperature), followed by aspiration of supernatant and resuspension of cells in fresh media. Cells were resuspended at a concentration of 200,000 cells/mL each time the media were replaced.

#### 4.3.1. THP-1 Cell Differentiation

THP-1 cells were plated in a 24-well plate (Cat. No. 0030722019, Eppendorf, Hamburg, Germany) in a volume of 0.5 mL at a concentration of 400,000 cells/mL to obtain 200,000 cells in each well. To generate monocyte-derived macrophages, THP-1 cells were cultured in media containing 25 nM (15.4 ng/mL) phorbol 12-myristate 13-acetate (PMA, Cat. No 194804, MP Biomedicals) for 72 h. PMA stocks were prepared at 400 µM using 100% dimethyl sulfoxide (DMSO) as a solvent (Cat. No. D2650, Sigma-Aldrich). After 72 h in PMA-media, the media were replaced, and cells were cultured for 24 h in fresh, non-PMA media. The same culture conditions of 5% CO_2_ at 37 °C in a humidified incubator were used during this protocol.

#### 4.3.2. *M. vaccae* NCTC 11659

*M. vaccae* NCTC 11659 was provided as a 10 mg/mL stock suspension in sterile BBS; strain National Collection of Type Cultures (NCTC) 11659, batch C079-ENG#1, provided by BioElpida (Lyon, France). After 24 h in fresh, non-PMA media, the THP-1 monocyte-derived macrophages were challenged with 10, 30, 100, or 300 
μg/mL concentrations of *M. vaccae* NCTC 11659 using sterile BBS to dilute from a 10 mg/mL stock (batch ENG#1, BioElpida, Lyon, France) to the desired concentration of *M. vaccae* NCTC 11659. Stocks were swirled each time before being pipetted into the wells to ensure the suspension was distributed equally. *M. vaccae* NCTC 11659 treatment consisted of adding 20 µL of either sterile BBS (vehicle) or 0.26, 0.78, 2.60, or 7.80 mg/mL *M. vaccae* NCTC 11659 stocks for the 10, 30, 100, or 300 µg/mL concentrations of *M. vaccae* NCTC 11659, respectively.

#### 4.3.3. LPS

After 24 h of incubation of THP-1 monocyte-derived macrophages with different concentrations of *M. vaccae* NCTC 11659 or sterile BBS vehicle, cells were then exposed to LPS or vehicle in the same media containing the bacteria and were cultured for an additional 24 h in a humidified incubator at 5% CO_2_ and 37 °C. Cells were cultured in the following: (1) *Experiment 1*, either a 0 (vehicle), or a 250 ng/mL concentration of LPS (*Escherichia coli* 0111:B4, Cat. No. L2630, Sigma-Aldrich); or (2) *Experiment 2*, either a 0 (RPMI 1640 vehicle), 0.5, 2.5, or 250 ng/mL concentration of LPS (*Escherichia coli* 0111:B4, Cat. No. L2630, Sigma-Aldrich). LPS was suspended in RPMI 1640 medium at an initial concentration of 1 mg/mL to make desired stock solutions for the 0.5, 2.5, and 250 ng/mL concentrations of LPS. The LPS challenge consisted of adding 5 µL of either RPMI 1640 vehicle or 0.0525, 0.2625, or 26.25 µg/mL LPS stocks for the 0 (vehicle), 0.5, 2.5, or 250 ng/mL concentrations of LPS, respectively. The 0.0525, 0.2625, and 26.25 µg/mL stocks for the 0.5, 2.5, and 250 ng/mL concentrations of LPS were prepared fresh from a 1 mg/mL stock of LPS right before the LPS challenge.

### 4.4. NanoString Gene Expression Analysis

The NanoString CodeSet used was the nCounter Human Inflammation v2 Panel (Item No. 115000072; XT-CSO-HIN2-12, NanoString Technologies, Seattle, WA, USA). The NanoString nCounter^®^ Inflammation Panel is a multiplex gene expression analysis platform that includes 255 genes that represent a broad range of pathways related to inflammation including apoptosis, epidermal growth factor (EGF) signaling, interleukin signaling, Ras signaling, T cell receptor signaling, and toll-like receptor signaling. The CodeSet included 249 inflammation-related genes and 6 housekeeping genes. A number of previous studies have used the nCounter Human Inflammation v2 Panel (e.g., [87,88,89,90,91,92,93]). Total RNA was extracted using QIAGEN RNeasy spin columns according to the manufacturer-issued instructions (Cat. No. 74104, QIAGEN, Germantown, MD, USA). Once dissolved in nuclease-free water, RNA was quantified using a NanoDrop One machine (Cat. No. ND-ONE-W, ThermoFisher Scientific, Madison, WI, USA), diluted to 10 ng/mL, and frozen at −80 °C before submitting samples to the Veterans Health Administration, Rocky Mountain Regional Veterans Affairs Medical Center (RMRVAMC) Core Equipment facility for further processing according to the manufacturer-issued instructions.

### 4.5. Real-Time RT-PCR

After centrifuging 24-well cell culture plates at 500× *g* for 5 min at 4 °C, supernatants were collected and total RNA was harvested from differentiated cells using a standard method of phenol–chloroform extraction [94]. Briefly, after collecting 500 µL of supernatant from each well, 0.5 mL of TRIzol^®^ (Cat. No. 15596-026, Invitrogen, Waltham, MA, USA) was added to each well and plates were frozen at −80 °C for 3 days until they were processed further. After thawing the plates and extracting the aqueous layer using chloroform separation, but before the RNA was precipitated with isopropanol, 10 µg of glycogen (Cat. No. AM9510, Invitrogen) was added to the aqueous layer to increase the total RNA yield from precipitation. Once dissolved in nuclease-free water, the RNA was quantified, and purity was assessed (A260/A280 ratio) using a NanoDrop One machine (Cat. No. ND-ONE-W, ThermoFisher Scientific). cDNA synthesis was conducted with SuperScript II^TM^ (Cat. No. 18064-014, Invitrogen) following manufacturer instructions using approximately 500 ng of the total RNA. Two 
μL of cDNA was used as template material. Real-time RT-PCR was conducted in duplicate using the CFX96 Touch Real-Time PCR Detection System (Cat. No. 1845097, Bio-Rad, Hercules, CA, USA) and SYBR Green master mix (Cat. No. 204145, QIAGEN, Hilden, Germany). All genes that were analyzed were normalized using the gene *ACTB,* encoding beta-actin. Real-time RT-PCR data were represented as a fold increase over the lowest amount of mRNA expressed for each gene using the delta-delta Ct method. Real-time RT-PCR was performed with an independent set of cells separate from those used for flow cytometry. Sample sizes for real-time RT-PCR are listed in Appendix A. Low sample sizes in the 0 ng/mL LPS condition were typically a result of PCR non-detects in the presence of low expression of target genes under BBS/RPMI 1640 vehicle conditions.

#### Primers

cDNA sequences were obtained from Genbank at the National Center for Biotechnology Information (NCBI; www.ncbi.nlm.nih.gov (accessed on 8 August 2019)). Primer sequences (Appendix A) were designed using the PrimerQuest^TM^ Tool from the Integrated DNA Technologies (IDT) website (https://www.idtdna.com/pages (accessed on 8 August 2019)). Sequence specificity was tested using the Basic Local Alignment Search Tool at NCBI [95]. Primers were obtained from IDT. Primer specificity was verified by melt curve analyses. All primers were designed to span exon/exon boundaries and thus exclude amplification of genomic DNA. 

### 4.6. Statistical Analysis

The data were analyzed as described below.

#### 4.6.1. NanoString nCounter^®^ Inflammation Panel

An analysis of reference genes in the NanoString platform revealed that treatment conditions altered the expression of all six reference genes in the nCounter Human Inflammation v2 Panel, resulting in inadequate normalization. As the “accuracy and reliability of gene expression results are dependent upon the proper normalization of the data against internal reference genes” [59,96], it was not possible to use the standard normalization methods within nSolver and ROSALIND ^®^. Instead, all normalization was done using Removing Unwanted Variation-III (RUV-III), specifically designed to address this issue in the NanoString nCounter gene expression assay [59]. RUV-III normalized data were imported into ROSALIND^®^. Normalized data were analyzed by ROSALIND^®^ (https://rosalind.bio/ (accessed on 15 November 2022)) with a HyperScale architecture developed by ROSALIND^®^, Inc. (San Diego, CA, USA). The limma R library [97] was used to calculate fold changes and *p*-values in ROSALIND^®^. The statistical software program R was used to make volcano plots using the ggplot2 and ggrepel packages.

#### 4.6.2. Real-Time RT-PCR

Real-time RT-PCR data were represented as either the relative gene expression (i.e., relative to the lowest expression of the gene of interest) or the ratio of one gene’s expression to another gene’s expression. Relative gene expression was calculated using the 2^−ΔΔCt^ method and the ratio of one gene to another gene was calculated by finding the Cq difference between the two genes and raising this value to the power of 2. For a detailed methodology of how mRNA expression ratios were calculated, see the Appendix A. Real-time RT-PCR data were analyzed using a two-way ANOVA followed by Dunnett’s test, if appropriate, at a two-tailed alpha level of 0.05 using a single pooled error term for Dunnett’s test. Specifically, in the presence of a main effect of *M. vaccae* NCTC 11659, planned pairwise comparisons between the BBS control condition and each concentration of *M. vaccae* NCTC 11659 (10 µg/mL, 30 µg/mL, 100 µg/mL, or 300 µg/mL) were conducted within the RPMI 1640 or LPS conditions (0.5 ng/mL, 2.5 ng/mL, or 250 ng/mL). In the presence of a main effect of LPS, planned pairwise comparisons between the RPMI 1640 control condition and each concentration of LPS (0.5 ng/mL, 2.5 ng/mL, or 250 ng/mL) were conducted within the BBS or *M. vaccae* NCTC 11659 conditions (10 µg/mL, 30 µg/mL, 100 µg/mL, or 300 µg/mL). In the presence of a significant interaction between *M. vaccae* NCTC 11659 and LPS, both types of planned pairwise comparisons were made.

#### 4.6.3. Software

Generating real time RT-PCR relative expression values and real time RT-PCR statistical analysis was conducted using Python (version no. 3.8.1). All Python code is available on GitHub at https://github.com/evho3333/Statistical-Analysis.git (accessed on 8 July 2022).

## Figures and Tables

**Figure 1 ijms-24-05176-f001:**
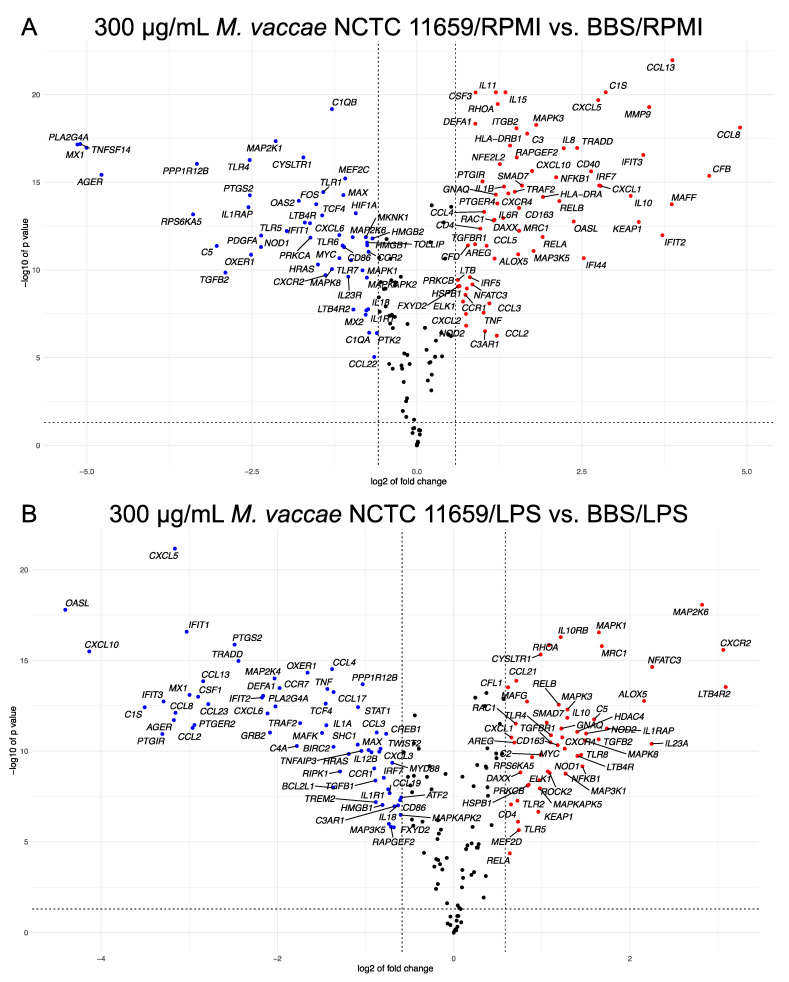
Volcano plots illustrating that (**A**) among THP-1-derived macrophages subsequently exposed to RPMI 1640 vehicle for 24 h, *M. vaccae* NCTC 11659 induced both proinflammatory and anti-inflammatory markers in human THP-1-derived macrophages and (**B**) among THP-1-derived macrophages subsequently challenged with 250 ng/mL lipopolysaccharide (LPS) for 24 h, *M. vaccae* NCTC 11659 shifted THP-1-derived macrophages toward an anti-inflammatory bias, for example, with increased *IL10*, *IL10RB*, *TGFB2, TGFBR1, MRC1* and decreased *IL12B* expression. The x-axis is the log base 2 of the fold change and the y-axis is the negative log base 10 of the *p*-value. Each dot in the figure represents a specific gene. The blue dots in the upper left quadrant of the volcano plot represent the genes that were expressed at lower levels in the (**A**) 300 μg/mL *M. vaccae* NCTC 11659/RPMI 1640 vehicle group or (**B**) 300 μg/mL *M. vaccae* NCTC 11659/LPS group relative to the BBS vehicle/RPMI 1640 vehicle group or BBS vehicle/LPS group, respectively. The red dots in the upper right quadrant of the volcano plot represent the genes that were expressed at higher levels in the (**A**) 300 μg/mL *M. vaccae* NCTC 11659/RPMI 1640 vehicle group or (**B**) 300 μg/mL *M. vaccae* NCTC 11659/LPS group relative to the BBS vehicle/RPMI 1640 vehicle group or BBS vehicle/LPS group, respectively. The black dots represent genes that were not differentially expressed between the groups (i.e., the absolute value of the log base 2 of fold change < 1.5 or *p*-value > 0.05). The dashed vertical lines represent log base 2-fold changes of approximately –0.6 or 0.6 (i.e., genes with an absolute fold change of at least 1.5). The horizontal dashed line is the negative log base 10 of the adjusted *p*-value alpha level of 0.05. The volcano plot was generated using the ggplot2 and ggrepel packages. See Appendix A. List of abbreviations for all gene symbols for a complete list of definitions of gene symbols (Appendix A).

**Figure 2 ijms-24-05176-f002:**
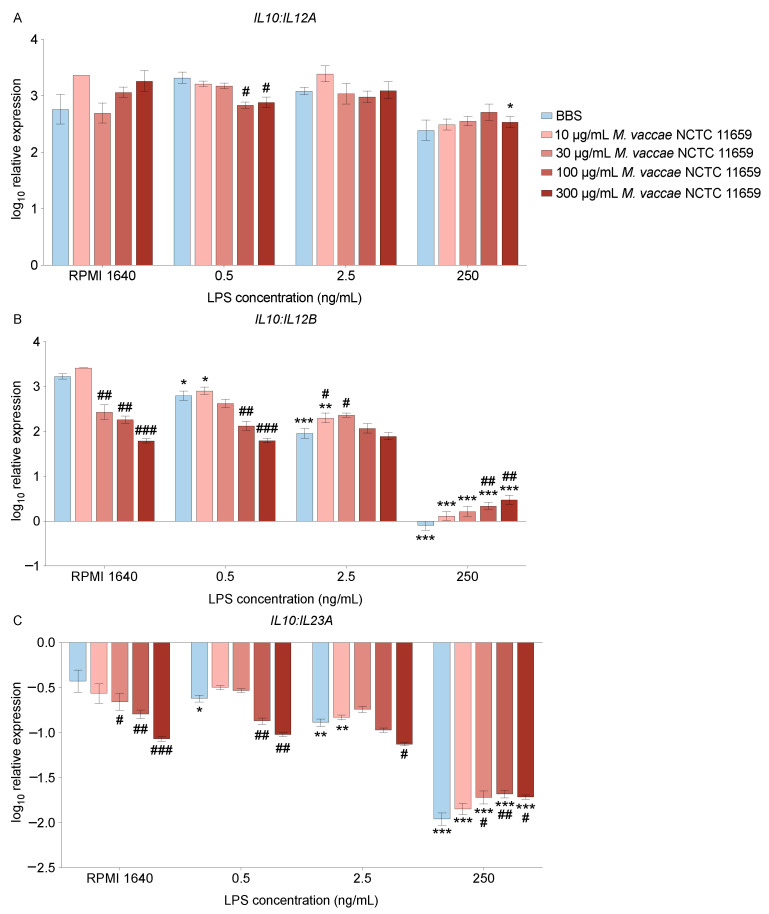
Heat-killed *M. vaccae* NCTC 11659 modulates lipopolysaccharide (LPS)-induced exaggeration of inflammation in THP-1 monocyte-derived macrophages by increasing the ratio of *IL10* mRNA expression to *IL12B* and *IL23A* mRNA expression, encoding the subunits of the heterodimeric cytokine, IL-23. Gene expression was measured using real-time reverse transcription polymerase chain reaction (RT-PCR) and is represented relative to the highest Cq value for each gene. Data represent means ± SEM. Numbers on the x-axis indicate the concentration of LPS in ng/mL. Data were analyzed using a two-way ANOVA followed by Dunnett’s multiple comparisons test using a single pooled error value from all 20 treatment groups, if appropriate, at a two-tailed alpha level of 0.05. Darker colors represent sequentially higher concentrations of *M. vaccae* NCTC 11659. * *p* < 0.05, ** *p* < 0.01, *** *p* < 0.001, effect of LPS, within the same *M. vaccae* NCTC 11659 condition, using the RPMI 1640 groups as the comparison groups. # *p* < 0.05, ## *p* < 0.01, ### *p* < 0.001, effect of *M. vaccae* NCTC 11659, within the same LPS condition, using the BBS condition as the comparison group. (**A**) *IL10:IL12A* mRNA expression ratio upon challenge with different concentrations of *M. vaccae* NCTC 11659 and LPS. (**B**) *IL10:IL12B* mRNA expression ratio upon challenge with different concentrations of *M. vaccae* NCTC 11659 and LPS. (**C**) *IL10:IL23A* mRNA expression ratio upon challenge with different concentrations of *M. vaccae* NCTC 11659 and LPS. Abbreviations: BBS, borate-buffered saline; LPS, lipopolysaccharide; NCTC, National Collection of Type Cultures. Sample sizes can be found in Appendix A.

**Figure 3 ijms-24-05176-f003:**
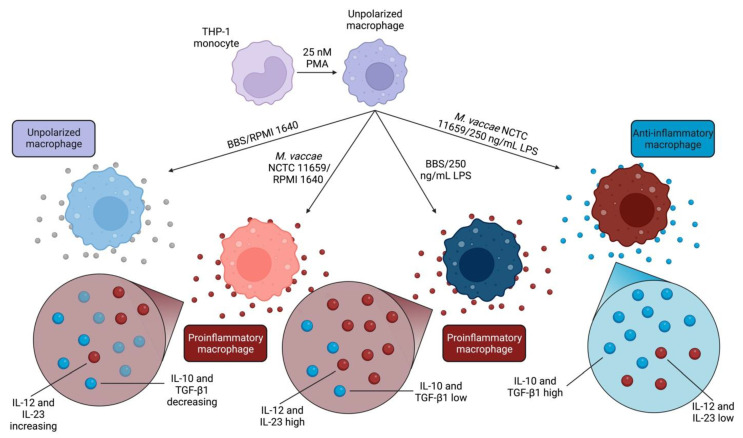
Graphical representation of the macrophage polarization phenotype of four treatment groups as described above. Proinflammatory polarization was observed in treatment groups challenged with either *M. vaccae* NCTC 11659 followed by RPMI 1640 vehicle or borate-buffered saline (BBS) vehicle followed by 250 ng/mL lipopolysaccharide (LPS) (as evidenced by *IL10:IL12B, IL10:IL23A*, *TGFB1:IL12B*, and *TGFB1:IL23A* ratios). An anti-inflammatory polarization was observed in the treatment groups challenged with *M. vaccae* NCTC 11659 followed by 250 ng/mL LPS. Abbreviations: IL-10, interleukin 10; IL-12, interleukin 12; IL-23, interleukin 23; LPS, lipopolysaccharide; PMA, phorbol 12-myristate 13-acetate; TGF-β1, transforming growth factor beta 1. Figure made with biorender.com.

## Data Availability

All data presented are available in the Appendix A.

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
