# Peer review of "Mycobacterium vaccae NCTC 11659, a Soil-Derived Bacterium with Stress Resilience Properties, Modulates the Proinflammatory Effects of LPS in Macrophages"

_ijms, 2023, doi:10.3390/ijms24065176_

Round 1

Reviewer 1 Report

The work by Holbrook and colleagues is oriented to show the immunoregulatory (anti-inflammatory) effects of some bacterial strains. This is placed in the context of the natural regulation of the inflammatory responses that can impact human health under ‘pro-septic’ environments. The idea is interesting and has been addressed in the past by several groups. However, as presented by the authors, the work has significant limitations in the way they run the assays and present the conclusions.
The first limitation is the use of a cell line (THP1) instead of monocytes and/or macrophages from healthy donors. This is an important limitation in the results presented in this work. THP1 exhibit only partially overlapping responses as human macrophages.

The second limitation is the modest effects (anti-inflammatory) observed in their analysis. All conclusions are based on a single time-point of the transcriptomic analysis. Why different kinetics in the transcriptional response is omitted? It is necessary to show that the release of cytokines, chemokines, and/or immunomodulatory molecules are in fact repressed and not only modified at a fixed time-point.

The third limitation is regarding the definition of pro- or anti-inflammatory of various molecules. Again, a time course is necessary for supporting these conclusions. Multiplexing assays are necessary to support the conclusions based only on transcriptional, cell-line limited studies.

Author Response

Please see uploaded document.

Reviewer 2 Report

The manuscript submitted by Dr. Holbrook et al. found that M. vaccae NCTC 11659 could inhibit the LPS-induced inflammation in THP-1 cells. The authors worked out this conclusion by in vitro study but described it at a super long length. There are numerous concerns to raise, and here I list some major concerns:

1. The conclusion can’t support the big title. The authors demonstrated the bacterium’s effect on inflammation only using cells. Besides, why did the authors connect the anti-inflammation effects to anti-stress-related psychiatric disorders? The authors have to perform in vivo experiment to evaluate this hypothesis.

2. The whole MS looks like an experimental note, not a scientific article:

a. The MS is too long and has too many repeat contents. Basic experimental techniques can be described briefly to save words, such as cell culture, total RNA extraction, or qRT-PCR. The authors are recommended to simplify the MS. For example, the M&M section can be simplified as 2.1. Experiment design, 2.2. M. vaccae NCTC 11659 preparation, 2.3. cell culture and bacterium treatment, 2.4. Nanostring gene expression analysis, 2.5. Real-time RT-PCR, 2.6. Statistical analysis.

b. The MS structure is messed up. For example, 2.4.1. section belongs to Results, 1st paragraph of 2.5. belongs to the Introduction section, etc. Too many same problems can’t raise here.

3. The MS performance is also needed to improve. For example, LPS could induce inflammation is well-known. So, 3.1.1. LPS and Figure 2 are meaningless and should be deleted. Fig. 2-5 for Nanostring analysis are also needed to rethink in a better way for better understanding.

Author Response

Please see uploaded document.

Reviewer 3 Report

Dear Authors,

The publication “ Mycobacterium vaccae NCTC 11659, a soil-derived bacterium with stress resilience properties, modulates the proinflamma[1]tory effects of LPS in macrophages: Implications for prevention and treatment of stress-related psychiatric disorders”  addresses a very important aspect of the body's response to chronic stress. I believe that the research is valuable and multi-level, and the article itself is refined. I recommended for publication in its present form.

Author Response

Please see uploaded document.

Round 2

Reviewer 1 Report

The authors revised the manuscript as suggested by the Reviewers. The present version clarifies many of the concerns I raised. However, the limitations regarding the model of choice still persist.

Reviewer 2 Report

Some concerns remained in the revised manuscript.

1. Although the authors deleted the subtitle from the title, the related contents in the Abstract and Introduction remained a lot. I understood that the authors wanted to connect M. vaccae NCTC 11659 to stress-related psychiatric disorders. Still, the evidence in this study only demonstrated its anti-proinflammation effects without any test on psychiatric disorders. Overemphasis is unnatural.

2. The Abstract section is too long. Please simplify it.

3. Figure 1 and Table 1 should be moved to supplementary materials.

4. There is no clear explanation of the underlying molecular mechanisms of M. vaccae NCTC on LPS-induced inflammation. Although the authors listed the DEGs (Figure 2), the relationships of these DEGs, the regulated signaling pathway, and networks should be mentioned—the same issue in 3.2.2.

5. 3.3 -3.5 section: I can’t understand the authors used seven pages (clean version) for the real-time validation experiment. I do not think the readers would be interested in it. We need to work on presenting the data clearly at a glance.

6. Again, the whole manuscript is still monotonous and lengthy without focus. The authors focused on recounting everything they’ve done with no good summary. It is not good to serve everything.

Round 3

Reviewer 2 Report

The authors addressed the issues well and can be accepted for publication.